# JUND plays a genome-wide role in the quiescent to contractile switch in the pregnant human myometrium

Nawrah Khader[ID][1], Anna Dorogin[2¤], Oksana Shynlova[2,3,4], Jennifer A. Mitchell[ID][1,5*]

1 Department of Cell and Systems Biology, University of Toronto, Toronto, Ontario, Canada, 2 Lunenfeld Tanenbaum Research Institute, Sinai Health System, Toronto, Ontario, Canada, 3 Department of Obstetrics and Gynaecology, University of Toronto, Toronto, Ontario, Canada, 4 Department of Physiology, University of Toronto, Toronto, Ontario, Canada, 5 Department of Laboratory Medicine and Pathobiology, University of Toronto, Toronto, Ontario, Canada

¤ Current address: Department of Cell and Systems Biology, University of Toronto, Toronto, Canada
* ja.mitchell@utoronto.ca

## Abstract

The myometrium, the muscular layer of the uterus, undergoes crucial transitions during pregnancy, maintaining quiescence throughout gestation, and generating coordinated contractions during labor. Dysregulation of this transition can lead to premature labor with serious complications for the infant. Despite extensive gene expression data available for varying myometrial states, the molecular mechanisms governing the increase in contraction-associated gene expression at labor onset remain unclear. Transcription factors, such as JUND and progesterone receptor (PR), play essential roles in regulating transcription of select myometrial contraction-associated genes, however, a broader understanding of their involvement in transcriptional regulation at a genome-wide scale is lacking. This study examines changes in transcription and JUND binding within human myometrial tissue during the transition from quiescence (term-not-in labor/TNIL) to contractility (term labor/TL). Total RNA-sequencing reveals a global increase in primary transcript levels at TL, with AP-1/JUND binding motifs overrepresented in the promoters of upregulated transcripts. Interestingly, ChIP-seq analysis demonstrates higher JUND enrichment in TNIL compared to TL tissues, suggesting its role in preparing the myometrium for labor onset. Integration of JUND and PR ChIP-seq data identifies over 10,000 gene promoters bound by both factors at TNIL and TL, including genes involved in labor-driving processes. Additionally, the study uncovers elevated levels of enhancer RNAs (eRNAs) at intergenic JUND peaks in laboring myometrial tissues, and implicates additional transcription factors, such as NFKB and ETS, in the regulatory switch from quiescence to contractility. In summary, this research enhances our understanding of the myometrial molecular regulatory network during pregnancy and labor, shedding light on the roles of JUND and PR in gene expression regulation

**Data availability statement:** All sequencing data is submitted to GEO (https://www.ncbi.nlm.nih.gov/geo/), accession GSE240816 and GSE244863.

**Funding:** This research was supported by funding from the Canadian Institutes of Health Research (https://cihr-irsc.gc.ca/), PJT-173252 held by JAM and OS. The funders had no role in the study design, data collection and analysis, decision to publish, or preparation of the manuscript.

**Competing interests:** The authors have declared that no competing interests exist.

genome-wide. These findings open avenues for further exploration, potentially leading to improved interventions for preventing premature labor and the associated complications.

## Author summary

The muscular layer of the uterus is responsible for the forceful contractions that deliver the baby at the end of pregnancy. During the majority of pregnancy this muscle is not able to contract, but before labor genes become expressed that allow the transition to a contractile state. Using human uterine muscle samples isolated during caesarean sections we determined the gene expression changes that occur and investigated regulatory proteins that bind to the genome to cause changes in gene expression. We identified a collaboration between the JUND transcription factor and progesterone receptor (PR) which together appear to regulate the expression of a subset of the genes expressed to allow the forceful contractions of labor. To further understand this process, we mapped genome-wide binding of JUND and found that JUND and PR co-occupy specific regulatory regions near genes involved in key labor-driving processes. These results suggest that JUND plays a central role in coordinating the transcriptional program that drives the switch from uterine quiescence to contractility. Our findings contribute to a better understanding of the molecular events that prepare the uterus for labor and may inform strategies to manage preterm birth.

## Introduction

The myometrium, the muscular layer of the uterus, plays a vital role throughout gestation in protecting the growing fetus by maintaining a non-contractile or quiescent state. At term, the myometrium is essential for generating the coordinated forceful contractions during labor required to deliver the fetus. Dysregulation in the signals that regulate the switch from the quiescent to contractile state can lead to premature labor or cause labor dystocia, resulting in the delivery of stillborn or underdeveloped infants at risk for complications later in life [1–3]. Despite the plethora of gene expression data generated from the myometrium at varying states, including labor at term compared to non-laboring pregnant myometrium [4–6], non-pregnant versus term pregnant [7], or depending upon the number of fetuses and timing of labor onset [8], we currently lack an understanding of the molecular mechanisms that govern the regulation of these gene expression changes.

Precise control of gene transcription is achieved by the collective action of proximal promoter elements and distal non-coding regulatory regions, such as enhancers, which mediate increased gene transcription. Transcription factors bind simple sequence motifs within both promoters and distal regulatory regions thereby regulating tissue-specific and signal responsive changes in gene expression. Multiple

transcription factors are responsible for regulating gene expression in a specific cellular context and these usually bind common regulatory elements as large complexes [9–13]. Whereas promoter regions are simple to identify, based on their location upstream of the transcript start site, enhancers can be located at megabase distances from the gene(s) they regulate [14–16], and are therefore challenging to identify. Existing methods to identify candidate enhancers involved in mediating the transcriptome include transcription factor binding, histone H3 acetylation on lysine residue 27 (H3K27ac), as well the production of non-coding enhancer RNAs (eRNAs) [17–20].

Current understanding of the transcriptional regulatory network changes in the myometrium that regulate the onset of labor remains incomplete. Previous studies have implicated the activator protein 1 (AP-1) family, and progesterone receptor (PR) transcription factors in regulating specific myometrial contraction-associated genes including: gap junction alpha 1 (*GJA1*), the oxytocin receptor (*OXTR*) and prostaglandin-endoperoxide synthase 2 (*PTGS2*) (reviewed in [21]). In rodent and human tissues, the AP-1 family member, Jun proto-oncogene D (JUND) is expressed at increased levels, and displays increased protein accumulation in the nuclei of laboring myometrial cells compared to non-laboring pregnant tissues [22,23]. Moreover, reporter assay experiments revealed enhanced expression from the *Gja1* promoter when members of the AP-1 subfamily FBJ osteosarcoma oncogene (FOS) and JUND were co-expressed in hamster myometrium cells [24]. Although progesterone bound to the PRB isoform of its receptor is generally thought to be involved in the repression of contractions during pregnancy, an interaction between the PRA isoform and AP-1 factors, in the absence of progesterone, was observed to cooperatively activate the expression of *Gja1* which is involved in myometrial contractions and labor timing [25,26]. Although these studies highlight the role of AP-1 factors and PRs at selected contraction-associated genes involved in labor onset, we lack information about the broader role these transcription factors play in the labor process.

To investigate the transition from quiescence to contractility in the myometrium, we examined changes in transcription and JUND binding within term human myometrium tissue collected during elective caesarean section before labor onset and compared it with human myometrium collected during active labor. Using total RNA-sequencing (RNA-seq), as opposed to the more widely used polyA selected RNA-seq approaches targeting processed mRNAs, we generated transcriptomic profiles which demonstrate a global increase in primary transcript levels in tissues obtained at term labor (TL), compared to non-laboring samples. The promoters of these upregulated transcripts contained an overrepresentation of AP-1 motifs, implicating AP-1 factors in the regulation of gene transcription changes at labor onset. Despite the observed increased transcription at labor, we observed increased biding of JUND in the ChIP-seq from non-laboring tissues compared to laboring tissues, suggesting a role for JUND in preparing the myometrium prior to labor onset. Interestingly, through integrating JUND and PR ChIP-seq, we found over 10,000 regions bound by both JUND and PR, at the promoters of genes, including many involved in labor-driving processes. Furthermore, we show that the increased transcription at labor is accompanied by higher levels of eRNA production at intergenic regions containing JUND binding and implicate additional transcription factors involved in the quiescence to contractility regulatory switch.

## Results

### Labor-associated genes are regulated by an increase in gene transcription at labor onset

To better understand the transcriptional changes that drive the transition from a relatively quiescent to contractile phenotype in the myometrium, we conducted total strand-specific RNA-seq of myometrial tissue obtained from non-laboring (TNIL, gestational age 37.6-39.1 weeks, n = 6), and laboring individuals at term (TL, gestational age 37.5-40.3 weeks, n = 6) [Fig 1A; see Methods]. Principal component analysis (PCA) of all samples revealed that replicates belonging to the TNIL group clustered together. For TL, 4 samples were clustered separately from the TNIL group; however, TL2 and 3 were clustered with the TNIL group, reflecting increased variability in the samples obtained in active labor (S1 Fig). Despite variability in the TL sample group, a larger number of genes were in the set that display increased expression at TL compared to the number of genes that exhibit higher expression at TNIL (Fig 1B). Differential gene expression analysis based on exon read counts uncovered a significant upregulation of 1,386 genes (log$_2$ fold change ≥ 1.5, $p$adj < 0.01) in the

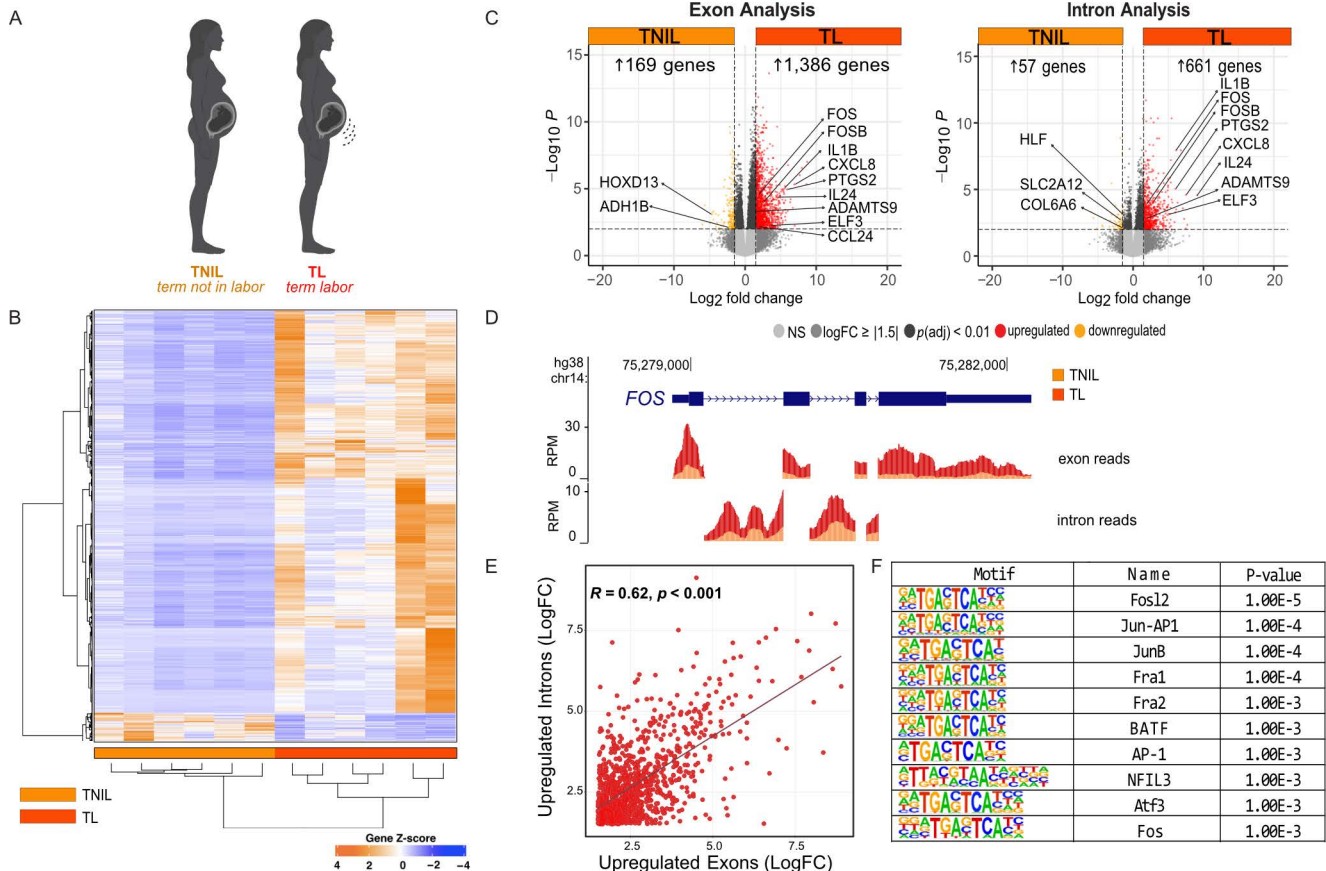

**Fig 1. Labor-driving genes are upregulated at the level of primary transcript production during labor.** A) Schematic showing the myometrium tissue samples used for RNA-seq that were obtained from patients at term not in labor (TNIL) and term labor (TL) [n = 6], created with BioRender.com. B) Heatmap of differential gene expression across all samples [abs log$_2$ fold change ≥ 1.5, *padj* ≤ 0.01]. C) Volcano plot showing differential gene expression between TNIL and TL [abs log$_2$ fold change ≥ 1.5, *p*adj ≤ 0.01], with exonic expression (left) and intronic expression (right) at TL compared to TNIL. D) Gene locus of labor-associated gene, *FOS*, which shows increased expression at TL (red) compared to TNIL (orange) at both the exon (top) and intron (bottom) level. E) Pearson correlation plot displaying the relationship between genes significantly upregulated during labor at the intronic versus the exonic level. F) Motif enrichment analysis at gene promoters significantly upregulated at the intron level.

TL myometrium in comparison to the TNIL myometrium ([Fig 1C](), *left and* [S1 Table]()). We compared our RNA-seq dataset to an expression analysis combining three different gene expression datasets, which found 126 genes that were upregulated in the TL myometrium compared to the TNIL myometrium [4,5,27]. From our differential gene expression analysis, we found 88 labor-upregulated genes were amongst the 126 found in agreement between the other three datasets, which include known labor-associated genes, prostaglandin G/H synthase 2 (*PTGS2),* and FOS like 1 (*FOSL1*)([S2 Table]()) [22,23,28,29]. Also part of this list are several immune cell chemoattractants and growth factors. Specifically, interleukin 6 (*IL6*), chemokine ligand 2 (*CXCL2*), and colony stimulating factor 3 (*CSF3*), are produced by human myometrial cells in culture and are thought to have a role in supporting immune cell infiltration of the myometrium, an important signal for labor onset [30,31]. S100 calcium binding protein A8 and 9 (*S100A8, S100A9*), however, are expressed in various immune cells at high levels [32] and may be present in this list due to the inclusion of immune cells in bulk tissue RNA-seq approaches.

To evaluate the role of gene transcription more specifically, we investigated significant changes in intronic read counts using a differential expression analysis (intron RNA-seq). Intron reads are a close proxy for primary transcript levels as

introns are rapidly co-transcriptionally spliced and degraded [33–36]. Analysis of expression changes at the intron-read level uncovered a significant elevation of intronic reads at 661 genes, including *FOS* (Fig 1C, right and S3 Table), in the TL compared to the TNIL human myometrium, suggesting that the transition to labor onset is marked by an increase in gene transcription. Similar to what was observed in the murine myometrium [37], *FOS* displayed enhanced exonic and intronic read counts in TL compared to TNIL myometrium, indicating an increase in *FOS* transcription occurs during this transition (Fig 1D). Comparative analysis identified a significant ($R = 0.62$; $p < 0.01$) positive correlation between upregulated genes at labor ($\log_2$ fold change ≥ 1.5) at the intron and exon level (Fig 1E). Collectively these data further support transcription as a driving force in regulating gene expression changes at labor onset in the human myometrium.

Next, we compared genes with increased intronic and exonic reads in our TL group to those upregulated specifically during labor in other human gene expression datasets, revealing a common set of 74 genes [4,5,27] [S2 Table]. To identify the transcription factors that may be involved in gene expression changes that underpin labor, we conducted a motif enrichment analysis using the promoters of these 74 genes. We uncovered a significant enrichment of motifs belonging to the AP-1 family (TGACTCA) [Fig 1F and S4 Table]. This observation is not surprising, as AP-1 dimers have long been implicated in regulating labor-associated gene expression in the myometrium. Specifically, AP-1 dimers exert transcriptional control over key labor genes, such as *Gja1*, *Ptgs2*, and *Oxtr* [21,24,37], which is further supported by the observed increase in nuclear JUND protein levels during labor in rodent and human myometrium [21,23]. Our analysis expands the role of AP-1 factors beyond specific genes and suggests their genome-wide involvement in regulating the increased expression of contraction-associated genes. In fact, when considering all 661 genes with increased intron levels in labor, AP-1 motifs remain significantly enriched in these gene promoters, along with NFKB (S4 Table). Together these findings indicate that similar to mouse myometrium [37], the contractility of human myometrium during TL is induced by a significant rise transcriptional activity, resulting in a rapid increase in primary transcript levels driven at least in part by AP-1 transcription factors.

### Increased JUND binding occurs in the non-laboring myometrium compared to the laboring myometrium

Since genes exhibiting increased expression levels during labor were associated with AP-1 motifs, and the AP-1 factor JUND is found at increased levels in human myometrial nuclei during labor [23] we investigated the JUND cistrome by conducting chromatin immunoprecipitation paired with sequencing (ChIP-seq) using human myometrial tissues (TNIL, n = 3 and TL, n = 2). Our initial peak analysis identified 70,455 JUND peaks in the TNIL samples and 40,327 JUND peaks in the TL samples with 89% of the TL peaks overlapping a TNIL peak (Fig 2A). 36,642 regions with peaks detected at TNIL did not display a significant peak at TL, suggesting a global loss of JUND binding occurs before labor onset (Fig 2A). To identify regions with significant differences in JUND association, differential peak analysis was conducted revealing a total of 6,740 peaks exhibiting significant differences between groups ($\log_2$ fold change ≥ |1|, $p$-value < 0.01). 6,546 TNIL peaks, display a significant loss at TL (referred to as TNIL JUND peaks) and 192 peaks show a significant gain at TL (referred to as TL JUND peaks, Fig 2B and S5 Table). The peaks that did not meet the threshold cut-off values for fold change and $p$-value were considered shared JUND peaks. Hierarchical clustering and PCA, as expected, demonstrated that ChIP-seq replicates from the same group cluster together, and the hierarchical clustering depicts differential JUND peaks between TNIL and TL samples, with more regions displaying increased binding at TNIL (shown in orange) compared to TL samples [Figs 2C, S2A and S2B].

Annotation of JUND peaks using the Genomic Regions Enrichment of Annotations Tool (GREAT) [38,39] reveals that TNIL-enriched peaks were associated with processes such as muscle hypertrophy, and TL-enriched peaks were linked to inflammation-related terms including positive regulation of immune system process (Fig 2D). Examples of specific genes linked to differentially bound JUND peaks are shown in Fig 2E. *ACTC1*, a cardiac muscle alpha actin encoding gene which is involved in muscle hypertrophy, and expressed at both TNIL and TL, displays increased JUND binding at TNIL (Fig 2E). At the *CCL24* gene, a labor-upregulated gene responsible for encoding a chemokine ligand that is involved in positive regulation of immune system processes, a TL enriched peak was observed (Fig 2E).

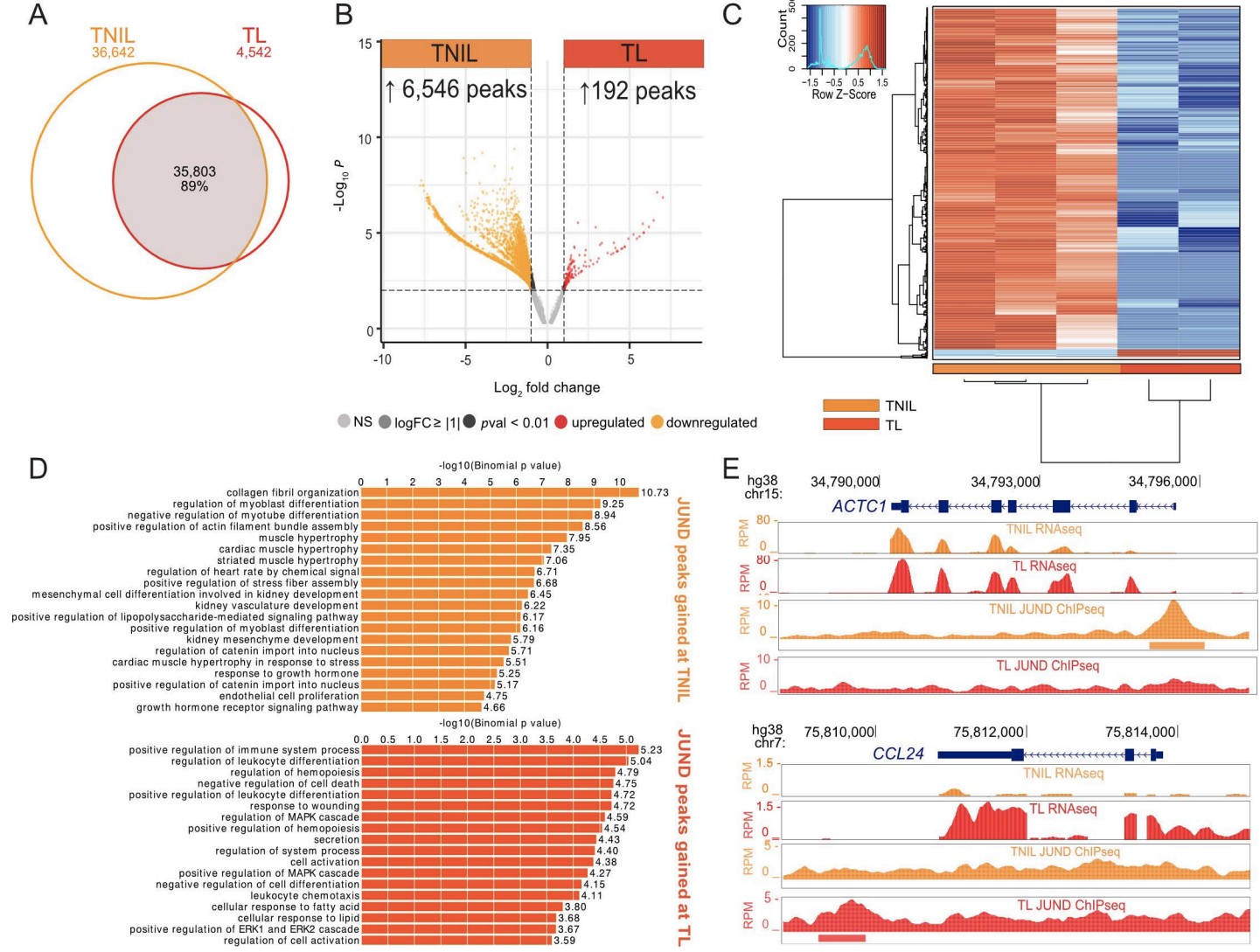

Fig 2. Differential genome-wide JUND binding reveals increased JUND binding in the term not in labor myometrium compared to the laboring myometrium. A) Venn diagram depicting overlapping JUND peaks obtained from ChIP-seq conducted from myometrium tissue samples at term not in labor (TNIL) and term labor (TL). B) Volcano plot shows differentially enriched regions [abs log₂ fold change ≥ 1, p-value ≤ 0.01] between TNIL and TL samples. C) Heatmap analysis shows the differentially bound regions [abs log₂ fold change ≥ 1, p-value ≤ 0.01] and the large subset of JUND enriched regions [red] at TNIL (n = 3) with lower enrichment [blue] at TL (n = 2). The histogram displays the relationship between read counts and colour. D) Bar graphs, ranked by binomial p-value, showing GO term biological processes associated with differential peaks from each timepoint that meet the GREAT cut-off (binomial FDR q value ≤0.05). E) JUND ChIP-seq peaks gained at either TNIL or TL mapped at ACTC1 gene locus and CCL24 gene locus, respectively.

Collectively, these results demonstrate that JUND may play distinct regulatory roles in the myometrium during pregnancy and labor, by modulating genes involved in muscle-related processes at TNIL and inflammation at TL, although the higher number of peaks at TNIL, and the number of shared peaks between TNIL and TL indicates that this transcription factor binds the genome before labor onset and is retained at these regions as labor is initiated and could therefore poise genes for later activation.

## JUND and PR co-bind promoters of labor-associated genes

Similar to JUND, progesterone receptors (PRs) have undergone extensive analysis for their regulatory roles in the myometrium [7,21,40]. Previous literature suggests a potential co-regulation of labor-driving genes, such as *GJA1*, by AP-1 factors and PRs, specifically the PRA isoform which is expressed at increased levels in the laboring myometrium [26]. To explore this association on a broader scale, we analyzed available ChIP-seq datasets of the total PR cistrome (antibody does not distinguish PRA and PRB) and the histone mark H3K27ac, associated with active regulatory regions, from TNIL and TL human myometrium (Fig 3A) [6]. These published datasets were compared with our JUND ChIP-seq to identify overlapping peaks. Notably, promoters of well-studied contraction-associated genes like *PTGS2* exhibited an H3K27ac signal and co-binding of JUND and PR, suggesting a potential necessity for the association of these two proteins in facilitating gene transcription (Fig 3B). Interestingly, although *PTGS2* was transcribed only in the TL myometrium, JUND, PR and H3K27ac were identified at the promoter at the TNIL stage, suggesting these features could be pre-marking labor associated genes and facilitating their rapid activation as the myometrium proceeds towards a contractile state.

To ascertain whether this pattern is more broadly observed, we investigated whether PR binds the same regions as JUND in either the TNIL, TL, or the set of shared JUND peaks. Enrichment plots display JUND binding at TNIL peaks is reduced or lost in TL samples, with shared JUND peaks displaying similar JUND binding at both timepoints, as expected (Fig 3C). Analysis of TNIL (Fig 3D; top row) or TL (Fig 3D; bottom row) JUND peaks indicated no co-enrichment of PR. Strikingly, JUND peaks enriched at TNIL, but not those enriched at TL, exhibited H3K27ac enrichment, suggesting that overall JUND peaks enriched at TL may be less functional in driving transcription (Fig 3D; right column). The shared set of JUND-bound regions displayed both high enrichment of PR and H3K27ac signatures, indicating that co-binding of PR and JUND occurs before labor onset and is maintained as the myometrium enters labor (Fig 3D; middle column). Interestingly, we noted a modest decrease in both JUND and PR binding at shared JUND peaks during labor compared to TNIL that was accompanied by an increase in H3K27ac signal (Fig 3D).

Noting that the shared JUND peaks contain the most PR co-association, we next conducted an overlapping analysis to assess the role of PR binding at these shared JUND peaks. In the overlapping analysis, we identified 12,081 regions (~25%) exhibiting overlap between shared JUND peaks and regions bound by PR (Fig 4A). While shared JUND or PR only peaks exhibited roughly 35–40% peaks in gene promoters (gene TSS +/- 1kB), 30% peaks in intronic regions, and 18% in intergenic peaks, the majority of the JUND and PR co-bound peaks were located within promoter regions (87.6%; Fig 4B). Previous studies have determined that PRs can bind DNA either through the canonical progesterone response element (PRE, 5′-GnACAnnnTGTnC-3) or through protein-protein interactions with AP-1 factors bound at AP-1 motifs. To investigate these possibilities, we looked at motif enrichment in the JUND, PR, or co-bound sets of peaks. Motif enrichment analysis highlighted a substantial enrichment of AP-1 motifs, in the shared JUND peaks without PR binding (Fig 4C). Within "PR-only" peaks, we noted an overrepresentation of nuclear receptor motifs (NR; HRE), including the androgen receptor element (ARE), which matches the PRE consensus [41]. Notably, there was little to no enrichment of PR-containing motifs [except for a related glucocorticoid receptor (GRE) motif ($p$-value 1e-4)] in the JUND and PR co-bound peaks (Fig 4C and S6 Table) and instead the predominant motifs are for AP-1 factors. These data suggest that when PR and JUND are co-bound, they associate with the genome largely using AP-1 motifs. It was previously reported that PRs interact with JUND at the AP-1 binding site in the mouse promoter of the labor-associated gene *Gja1*, suggesting that both proteins are involved in modulating *Gja1* expression [26]. Consequently, our findings extend this mechanism to a genome-wide context revealing that most of the JUND/PR co-binding occurs at AP-1 motifs in the absence of PRE motifs.

Next, we examined which genes are associated with PR and JUND co-binding, and their expression at the transition to labor. To address this, we used our RNA-seq and overlapped promoters bound by both JUND and PR to discern whether they preferentially bind genes expressed in labor. From the initial set of 9,362 genes linked to co-bound peaks, we identified the overlap with genes significantly upregulated ($\log_2$ fold change ≥ 1.5, $p$adj < 0.01) or downregulated ($\log_2$ fold change ≤ 1.5, $p$adj < 0.01) in our RNA-seq data. Using a hypergeometric analysis, we found a significant enrichment

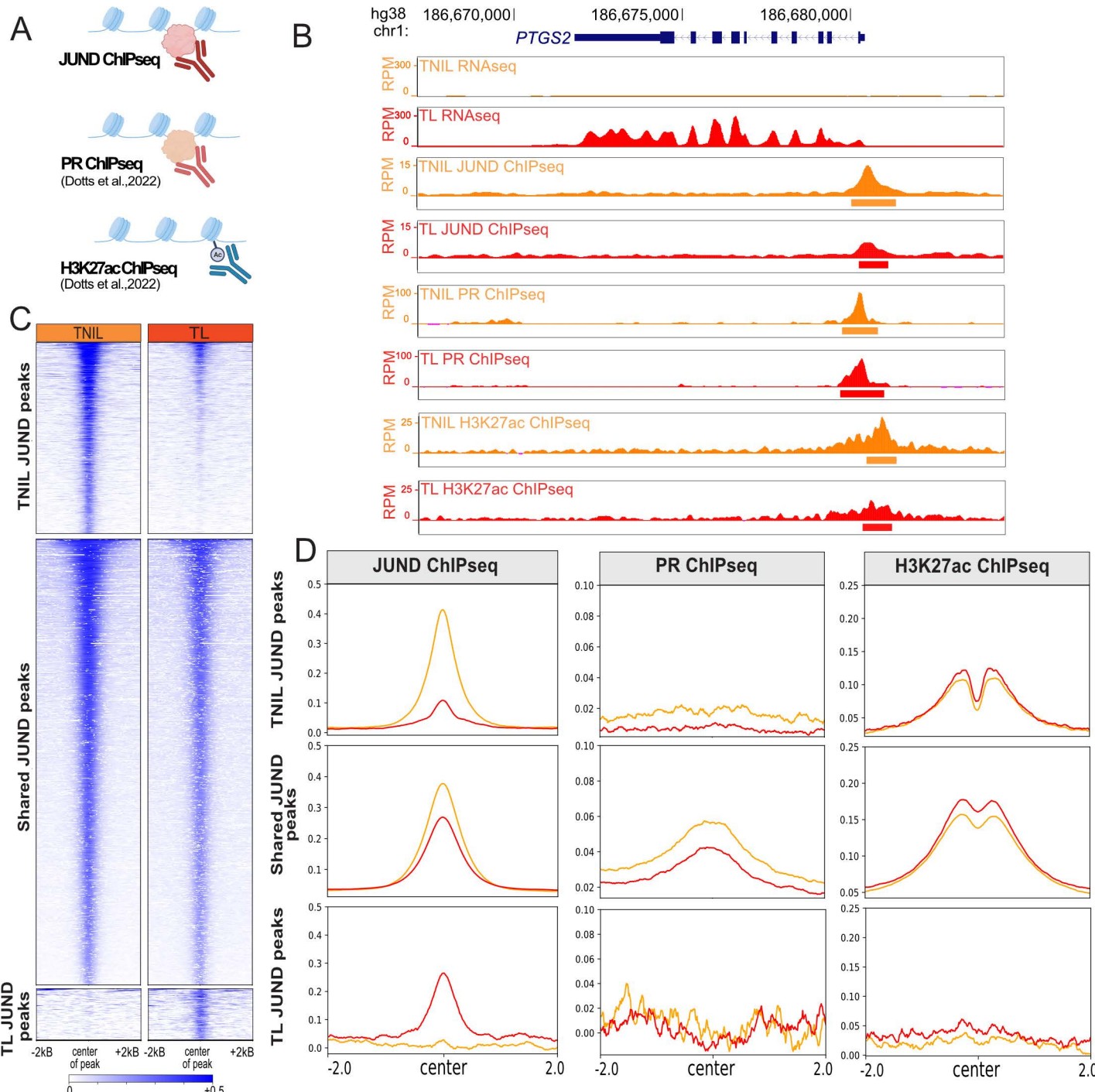

**Fig 3. Integrative analysis of JUND ChIP-seq with progesterone receptor (PR) and H3K27ac ChIP-seq demonstrates co-binding at promoters of genes involved in labor processes.** A) Schematic showing the ChIP-seq datasets used for analysis, created with BioRender.com. B) Human gene locus of a prominently studied labor-associated gene encoding, *PTGS2,* with RNA-seq, JUND ChIP-seq, PR ChIP-seq, and H3K27ac ChIP-seq signals exhibited during TNIL (orange) and TL (red). C) Enrichment heatmap plots of JUND ChIPseq at three clusters: differential regions bound by JUND at TNIL (*top*), regions bound by JUND at TNIL and TL (*middle*), and differential regions bound by JUND at TL (*bottom*). D) Profile plots showing JUND, PR, and H3K27ac ChIP-seq signals in reads per million (RPM) centered on peaks that are differentially enriched at TNIL (top row), JUND peaks shared amongst TNIL and TL (middle row), and significantly enriched at TL (bottom row). TNIL data displayed in orange and TL data displayed in red.

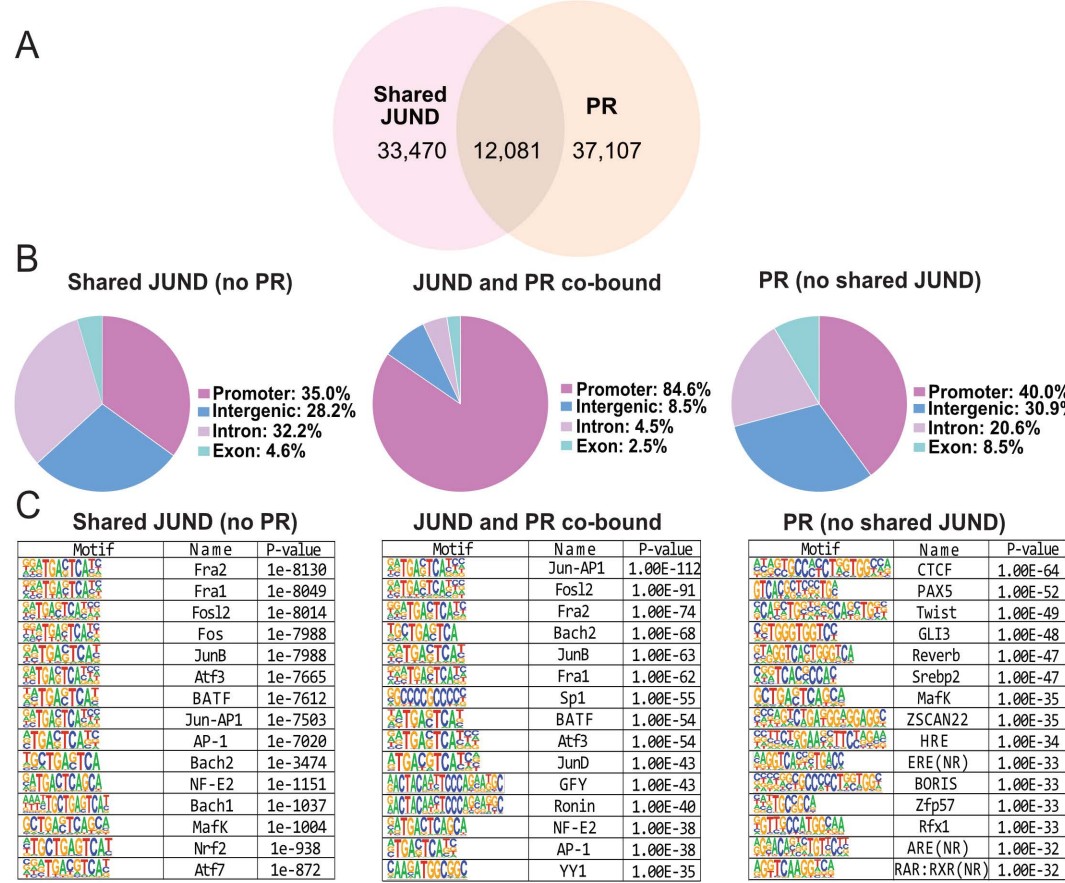

**Fig 4. Progesterone receptor (PR) and JUND co-bind at AP-1 motif containing gene promoters.** A) Venn diagram showing the overlap of JUND regions that are shared amongst TNIL and TL, with regions bound by PR. B) Functional genomic annotation of peaks bound by shared JUND peaks (*left*), co-bound by JUND and PR (*middle*) or PR (*right*). C) Motif analysis is displayed for peaks bound by shared JUND peaks (*left*), co-bound by JUND and PR (*middle*) or PR (*right*).

($p$<1.357e-32) of shared JUND and PR co-bound promoters in upregulated genes at labor (442 genes) and only a minimal overlap to downregulated genes (6 genes, S7 Table). Gene ontology analysis revealed an enrichment in biological processes related to inflammation, cell communication, and regulation of cytokine production, essential processes for labor in the set of JUND/PR co-bound and upregulated genes (S8 Table). This analysis suggests that JUND and PR, already co-bound to several gene promoters in the TNIL myometrium, jointly modulate the transcription of labor-associated genes, particularly those involved in inflammation and contraction, underlying the myometrial transition to labor.

**Active labor is associated with increased enhancer RNA production at acetylated intergenic regions bound by JUND**

Having determined that JUND and PR play a prominent role at labor-associated gene promoters prior to the onset of labor, we investigated potential triggers causing the surge in transcription observed at labor onset. Since enhancers are critical players in transcriptional state changes during development, we next investigated the set of intergenic regions bound by JUND that overlap the active enhancer associated feature, H3K27ac. Overlapping analysis uncovered 23,363 shared JUND and H3K27ac peaks at TL (Fig 5A), which were filtered (by removing regions that overlap a gene) to retain

PLOS Genetics

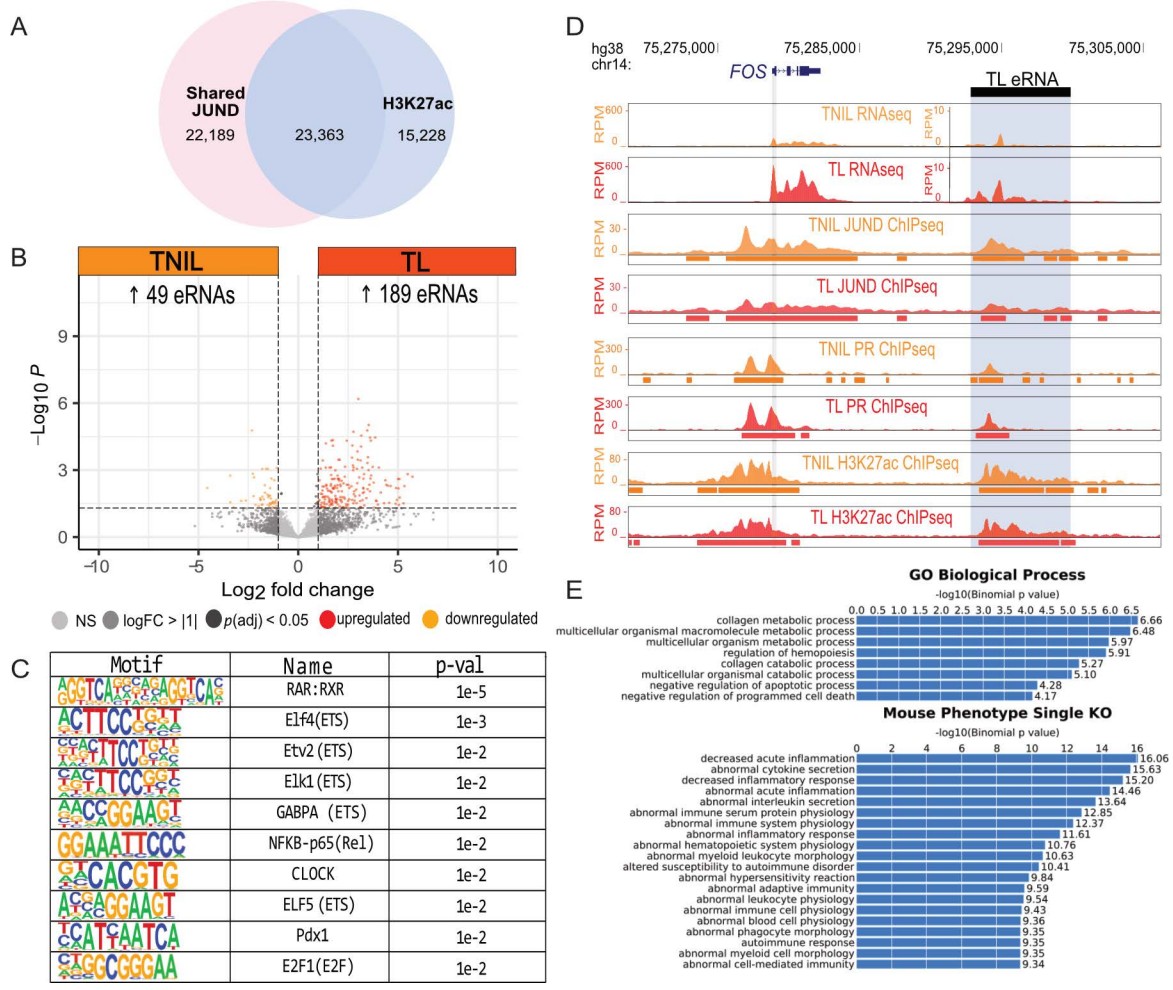

**Fig 5. H3K27ac intergenic regions bound by JUND demonstrate a significant increase in enhancer RNA (eRNA) production at labor and are associated with candidate labor-driving transcription factor motifs.** A) Venn diagram showing overlap of shared JUND peaks with H3K27ac peaks at TL. B) Volcano plot shows the differential eRNA expression between TNIL and TL (abs log$_2$ fold change ≥ 1, $p$-value ≤ 0.05), with higher eRNA levels at TL. C) Motifs belonging to the top ten transcription factors that are enriched at acetylated intergenic regions bound by JUND that exhibit a significant increase in eRNA at TL. D) Profile of region downstream of the labor-associated gene, *FOS*, shows significantly increased eRNA levels at TL compared to TNIL. E) Gene ontology analysis for genes associated with acetylated intergenic peaks bound by JUND.

6,457 intergenic peaks to identify candidate enhancer regions. In addition to containing the H3K27ac modification, transcriptional enhancers can give rise to non-coding eRNAs. Since JUND and H3K27ac are present in both TNIL and TL samples we investigated to what extent eRNA production was dynamic at these regions. Differential expression analysis of JUND/H3K27ac regions revealed an upregulation of 189 eRNAs at TL compared to TNIL (log$_2$ fold change ≥ 1, $p$adj < 0.05, Fig 5B and S9 Table). We hypothesized that eRNA production at these regions is increased by new transcription factors that co-bind with JUND and are required to increase gene transcription at labor onset. To identify candidates that may be important in this context, we conducted motif enrichment analysis on the JUND-bound intergenic regions, displaying H3K27ac and increased eRNA in labor (Fig 5C and S10 Table). Notably, there is a pronounced overrepresentation of motifs that bind factors within the Erythroblast Transformation Specific (ETS) family, including ELF4, ETV2, GABPA,

ELF5, and ETS1. Besides ETS family members, the enrichment analysis also included NFKB-p65 (RELA), a mediator of the pro-inflammatory cascade at labor onset (reviewed in [21,42]).

Among the eRNAs that exhibited significant upregulation during labor, we observed a non-coding region downstream of the *FOS* gene that includes prominent PR and JUND peaks as well as the H3K27ac signature (Fig 5D). Through lift over of this region from the human genome (hg38) to the mouse genome (mm10), we identified a comparable non-coding region downstream of the mouse *Fos* gene with known H3K27ac, RNA polymerase II association, and upregulation of eRNA in the laboring myometrium [37]. These data suggest the existence of a conserved enhancer capable of regulating *Fos* transcription in the laboring mouse and human myometrium. Further association of the regions with increased eRNA production to nearby genes identified enrichment in biological processes such as regulation of collagen production and genes linked to inflammatory phenotypes in mouse knockout studies (Fig 5E), as well as a significant enrichment of genes upregulated at TL (p < 3.756e-25, hypergeometric test). Collectively, these findings lead us to propose that the expression of labor-associated genes is governed by transcriptional regulatory mechanisms linked to JUND binding at both promoters and enhancers.

## Discussion

Based on our analysis of the JUND cistrome, we propose a model illustrating the molecular changes in the myometrium during the transition from quiescence to a laboring state (Fig 6). At late gestation, JUND binding occurs at labor-associated gene promoters and distal enhancers, marking regulatory regions for activation at labor onset. The observed co-binding of PR and JUND at labor-associated gene promoters occurs at AP-1 motifs and based on additional evidence, as described below, we propose that JUND homodimers at promoters and enhancers are replaced with JUND:FOS family heterodimers at the transition to labor. Concurrently, as the myometrium transitions to labor, distal regulatory regions containing prominent JUND binding and H3K27ac signatures recruit additional transcription factors, through NFKB and ETS motifs. This larger complex of transcription factors at the labor-associated enhancers causes the production of eRNA and increased transcription of target labor-associated genes (e.g., *PTGS2, FOS,* and *CXCL8*). In support of the NFKB role in labor onset, an increase in active phosphorylated RELA and elevated levels of RELA-p50 heterodimers were detected in the nuclei of human myometrial cells from laboring pregnant tissues [43,44], although the regions of the genome bound by these factors in the myometrium have not been determined.

The observed increase in myometrial primary transcript levels, associated with AP-1 motif enrichment at gene promoters aligns with prior research in the mouse myometrium also suggesting that AP-1 factors facilitate enhanced transcriptional activity during the transition from the quiescent to the contractile phenotype [37]. Past studies supporting this concept delved into labor-associated gene activation at a single-gene level; for example, mutation in an AP-1 binding site within a synthetic *Gja1* gene promoter altered AP-1-mediated reporter gene expression [24]. This functional involvement likely extends beyond the *Gja1* promoter, as evidenced by AP-1 motifs and JUND binding from our analysis of other labor-associated gene promoters like *OXTR*, *CXCL8*, and *PTGS2* [45,46]. Building on studies at the *Gja1* gene locus, it has been theorized that during gestation, JUN:JUN homodimers initially bind to the regulatory regions of the quiescent myometrial genome. Subsequently, these homodimers are replaced by JUN:FOS heterodimers, activating gene transcription for contractility, as JUN:FOS heterodimers have higher transactivation capabilities [21–24]. Consistent with this theory, our RNA-seq reveals elevated expression of *FOS*, *FOSB*, and *FOSL1*, and relatively stable levels of *JUN* and *JUND* RNA in human TL tissues compared to TNIL, similar to observations in mouse laboring tissues [37]. Interestingly, quantitative comparison of shared JUND peaks at TNIL and TL revealed a global decrease in JUND occupancy, coincident with an increase in H3K27ac modification at these regions. These findings could be interpreted as the formation of JUND:JUND homodimers at TNIL, which are replaced by JUND:FOS heterodimers at TL, causing a decrease in JUND binding but not a complete loss of the peak. The observed increase in H3K27ac could be a reflection of the increased transactivation potential of the JUND:FOS heterodimer complexes. A limitation of this study is our inability to conduct ChIP-seq using

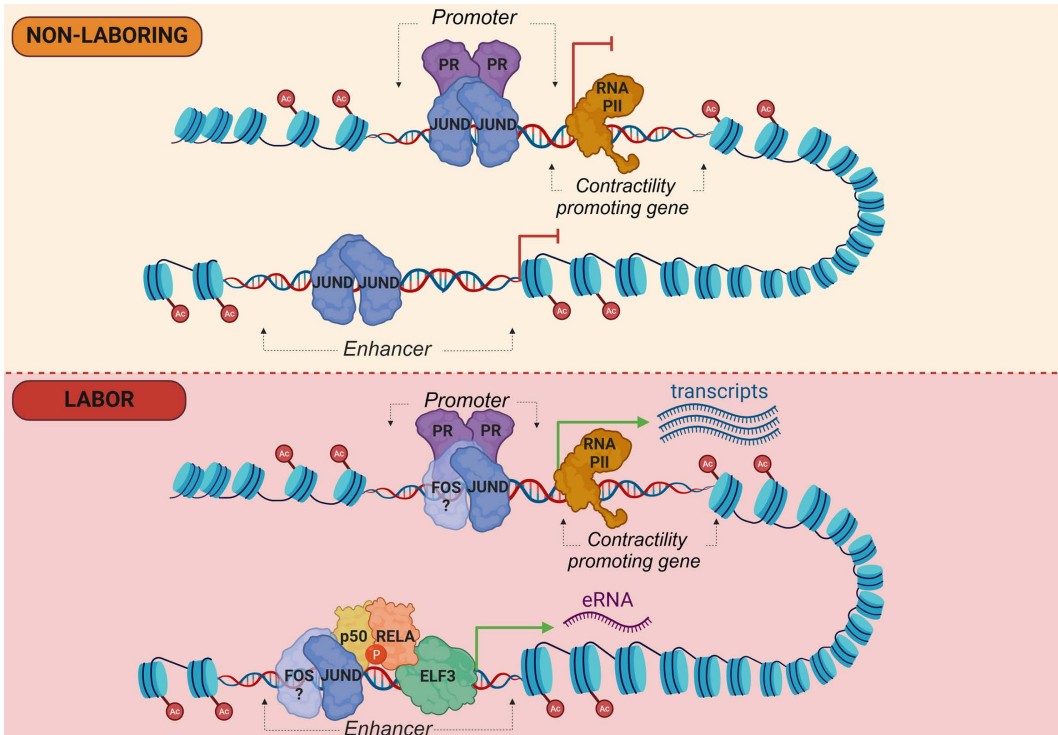

**Fig 6. Model of the human myometrial chromatin landscape at labor associated genes prior to and during the onset of labor.** At the non-laboring stage (top), labor associated gene promoters and enhancers display active histone modifications (H3K27ac) and associate with JUND homodimers. This association, along with the H3K27ac signature and PR recruited by AP-1 proteins, may be essential in priming these genes for later activation. At labor (bottom), JUND homodimers are replaced by JUND:FOS family heterodimers, due to increased FOS family protein abundance at labor. At intergenic enhancers, motifs belonging to candidate regulatory transcription factors from families that include ETS (ELF3) and NFKB -p65 (RELA) recruit these additional factors and the larger, more active, complex leads to eRNA production and gene transcription that shifts the myometrium to a contractile state. Created with BioRender.com.

anti-FOS protein antibodies, likely due to the structural nature of the myometrial tissue which is a challenge for chromatin analysis. In the future, analyzing differences in the FOS cistrome between TNIL and TL myometrium, along with the JUND ChIP-seq dataset, would establish if the lower JUND enrichment in the TL myometrium tissues is due to a switch in AP-1 dimer composition.

Previous ChIP-seq experiments aimed at identifying PR-occupied regions in the myometrium revealed a significant enrichment of AP-1 motifs, suggesting an interaction between these factors in the myometrium [6,7]. Chromatin capture assays focusing on enhancers driving PR-regulated genes in the myometrium also indicated an overrepresentation of AP-1 motifs, reinforcing the genome-wide regulatory role of AP-1 in myometrial function [47]. This AP-1-PR interaction has been previously postulated in breast cancer cells where PR-occupied regions are similarly enriched in AP-1 motifs [48]. A functional PR and AP-1 interaction was demonstrated at the *Gja1* promoter as enhanced reporter expression was observed when the PRA isoform was combined with JUN:FOS heterodimers in the absence of progesterone [26]. Our integrative analysis of JUND and PR bound regions confirms their interaction occurs primarily at promoters of labor-associated genes containing AP-1 motifs. In this context, the absence of PRE-containing motifs at these regions alludes to PR's indirect DNA binding, versus the regions bound only by PR where an enrichment of the PRE consensus suggests more direct DNA binding occurs. Although the available PR ChIP-seq data does not distinguish between PR isoforms we propose that the activation of labor-associated genes bound by JUND and PR involves unliganded PRA in this activating

complex, similar to what was observed at the *Gja1* promoter [26]. If true, this would further support the role for PRA and AP-1 factors as activators upon the functional withdrawal of progesterone in the human myometrium which is thought to be achieved by local progesterone metabolizing enzymes expressed in the human myometrium at increased levels during labor [26].

Examination of H3K27ac-enriched intergenic regions that give rise to non-coding RNAs identified a set of candidate labor-specific enhancer regions in the human myometrium. Importantly, here we demonstrated an increase in non-coding RNAs at acetylated intergenic regions in human laboring myometrium and identified a conserved candidate enhancer region downstream of the *FOS* gene that acquires eRNA in both human and murine laboring myometrium. As sequence conservation between mouse and human non-coding regions together with conserved chromatin features can be used to identify active enhancers [49–51], the conserved labor-specific enhancer features at this region make it an intriguing candidate enhancer that could drive increased *FOS* expression during labor.

The formation of protein complexes and recruitment of additional co-factors at distal regulatory regions that drive the myometrium to a contractile state can be better understood by assessing the binding activity of transcription factors at these regions. Motif analysis at the JUND-bound regions revealed RELA, from the NFkB family, as a transcription factor that may be involved in the labor associated transcriptional complex. Enrichment of RELA motifs at these regions aligns with its known pro-labor inflammatory role (reviewed in [21]). For instance, promoters of contraction-associated genes in the laboring mouse myometrium exhibited an overrepresentation of RELA along with AP-1 motifs, suggesting these factors may interact at regulatory regions to drive gene expression changes in the myometrium [37]. This interaction between AP-1 and RELA has previously been postulated and shown to regulate pro-labor inflammatory genes like *IL-8* in myometrial cells [44,46].These findings highlight the potential of RELA as an interaction partner for the AP-1 factors and a transcriptional activator of gene expression in controlling labor initiation. Our motif enrichment analysis at these potential enhancer regions also revealed a significant overrepresentation of ETS factors, which is unsurprising since *ELF3* was shown to be significantly upregulated in both the human and mouse laboring myometrium [37,52]. *Elf3* was also upregulated in the myometrium of mifepristone-induced progesterone withdrawal preterm laboring mice compared to the non-laboring control animals [53] suggesting a connection between PRs and ELF3. Collaboration of ETS factors with PR is further supported by the presence of ETS motifs in PR bound regions of the laboring myometrium [6]. ELF3 is an interaction partner with NFKB in prostate cancer cells in response to pro-inflammatory stimuli, a pathway that also impacts myometrial cells [54]. ELF3 has also been shown to decrease activation of labor-associated collagen genes as well as AP-1 heterodimer mediated activation on the *Mmp13* promoter in chondrocyte cells [55,56]. Importantly, within myometrial cells, reporter assays have shown that ELF3 can decrease JUND: FOSL2 mediated activation of synthetic *Gja1* and *Fos* promoters, shedding light into the repressive regulatory role that ELF3 may exert, in concert with additional binding partners, on labor-associated genes [53]. With ELF3 playing a seemingly repressive role on the transcription of labor-associated genes, this raises the question of why these motifs are enriched in potential enhancer regions that drive the myometrial transition to labor. Based on its repressive role in other cells, we hypothesise that ELF3 motif enrichment in these enhancer regions suggests a fine-tuning role in inhibiting the activation effects of AP-1 and RELA, which may contribute to the timely completion of the labor-related transcriptional program following delivery of the fetus. Further investigation through chromatin immunoprecipitation experiments and reporter assays is warranted to elucidate the specific roles and functional interplay between ELF3, RELA, and AP-1 in the myometrium.

Altogether, our current findings provide insights into the molecular changes occurring at the chromatin and transcription levels, facilitating the myometrium's transition between the quiescent and contractile states. While prior studies have suggested a role for AP-1 factors, like JUND, in regulating the transcription of specific contraction-associated genes, the current study reveals JUND's broader involvement in preparing the myometrium for the large-scale transcriptional events at the transition to labor. Additionally, we have identified PR, RELA, and ELF3, as potential interaction partners to JUND on a genome-scale to fine-tune transcriptional changes crucial for the successful initiation and completion of labor. The

distinct epigenetic molecular profiles between the non-laboring and laboring myometrial states underscore the importance of investigating this transcriptional regulatory network more extensively.

## Methods

### Ethics statement

This study was carried out in accordance with the protocol approved by the Research Ethics Board, Sinai Health System REB# 02-0061A and REB# 18-0168A. All subjects donating myometrial biopsies for research gave written informed consent in accordance with the Declaration of Helsinki. All research using human tissues was performed in a class II certified laboratory by qualified staff trained in biological and chemical safety protocols and in accordance with Health Canada guidelines and regulations.

### Tissue collection

After receiving written consent, myometrial biopsies from healthy individuals undergoing elective Caesarean sections (term not in labor/ TNIL, n = 6) or Caesarean sections during term labor (TL, n = 6) were collected and transferred from the operating theatre to the laboratory. For the TL group, biopsies were obtained during emergency Caesarean sections due to breech presentation or fetal distress. Patients with preterm premature rupture of the membranes, clinical chorioamnionitis, fetal anomalies, gestational diabetes/hypertension, cervical cerclage, preeclampsia, antepartum hemorrhage, and autoimmune disorders were excluded from the study. After the delivery of fetus and placenta, a small 1.5 cm sliver of myometrium was collected from the upper margin of the uterine incision made in the lower uterine segment in both laboring and non-laboring individuals.

### Chromatin immunoprecipitation assay for JUND binding

Chromatin immunoprecipitation was performed as described previously [57] with minor modifications. Briefly, 200 mg of flash frozen human myometrium tissues obtained from patients at TNIL (n = 3) or TL (n = 2) were crushed and subjected to a secondary cross-linker, disuccinimidyl glutarate (DSG), for 30 mins at room temperature, prior to formaldehyde fixation using 1% formaldehyde for ten minutes. This was followed by quenching through adding glycine to a final concentration of 125 mM for 15 min with rotation at room temperature. Cell pellets were washed twice with ice-cold PBS followed by resuspension in ice-cold lysis buffer 1 (50 mM HEPES-KOH, 140 mM NaCl, 1 mM EDTA, 10% glycerol, 0.5% NP-40, 0.25% Triton X-100) for 10 min at 4°C with rotation. Lysates were centrifuged at 2000g for 5 min at 4°C before resuspending in ice-cold lysis buffer 2 (10 mM TrisHCl, 200 mM NaCl, 1 mM EDTA, 0.5 mM EGTA) for 10 min at 4°C. Nuclei were pelleted at 2000g for 5 min at 4°C before resuspending in ice-cold lysis buffer 3 (10 mM Tris-HCl, 100 mM NaCl, 1 mM EDTA, 0.5 mM EGTA, 0.1% Na-deoxycholate, 0.5% N-laurylsarcosine). Sonication was performed using a probe sonicator at 20 Amps (15 sec on/30 sec off) for 2 min at 4°C. After cell lysis and sonication, 300 µl 10% Triton X-100 was added to the sonicated lysate to precipitate any debris. Fifty microliters of cell lysate were retained as whole-cell extract (WCE), and the remaining lysate was split between two different immunoprecipitations. Chromatin lysates were incubated overnight with 5 µg of JunD (Abcam; ab181615) at 4°C with rotation. Protein A magnetic Dynabeads (80 µL, Invitrogen) were washed with 1% BSA/PBS before adding to the antibody-bound chromatin lysates and incubated overnight at 4°C with rotation. Lysates were washed four times with 1 mL of room temperature RIPA buffer and then with a TBS buffer. Antibody-bound lysates and WCE were eluted in an SDS-based elution buffer (50 mM Tris-HCl, 10 mM EDTA, 1% SDS) for 30 min at 65°C before addition of proteinase K and overnight reverse cross-linking at 65°C with shaking at 900 RPM. ChIP DNA was purified using the phenol/ chloroform method and eluted in 65 µL of nuclease-free water. Aliquots from the sonicated samples were run on a 1% agarose gel to confirm that chromatin was sheared to a range of 300–1000 bp in size. Enrichment was determined through

qPCR using the input control a to create a standard curve and extrapolate concentration values to the IP samples and calculate fold enrichment over negative regions, with triplicate technical replicates.

## ChIP-seq data analysis

ChIP samples were submitted for paired-end 150 bp read sequencing using standard Illumina HiSeq 2500 protocols. Raw fastq reads were quality assessed and trimmed using fastp [58] and mapped to the hg38 reference genome using STAR [59]. ChIP-seq peaks were called, relative to the input, for each individual replicate using MACS2 [60] with default parameters (p ≤ 0.01). Significantly conserved peaks between biological replicates were combined using irreproducible discovery rate (IDR), a statistical method that ranks peaks by signal strength across replicates, using a bimodal model to remove less reproducible peaks [61]. This ChIP-seq data analysis pipeline was applied to published ChIP-seq datasets (progesterone receptor [PR] and histone acetylation [H3K27ac] from [6]). Differential peak analysis was performed on MACS2 peaks using the DiffBind package [62]. Peaks with an absolute $\log_2$ fold change ≥ 1.5 and adjusted *p*-value ≤ 0.01 were considered significantly different between various timepoint comparisons. The GREAT v3.0 API [38] was used to analyze gene associations of differential JUND ChIP-seq peaks at both timepoints with parameters set to 5 kb upstream and 1 kb downstream from the gene TSSs (basal), or up to 100 kB (distal), to define gene-regulatory domains. Functional annotations of genes from GO biological processes were used to derive significant gene and function enrichments (binomial FDR q value of ≤0.05). Heatmaps and profile plots were generated using "computeMatrix" and "plotProfile" from deep-Tools. HOMER was used to find overlapping peaks between shared JUND regions and PR bound regions (mergePeaks. pl) and to perform motif analysis of the resulting cobound peaks (findMotifsGenome.pl) [10]. ChIPseeker [63] was used to functionally annotate peaks to genomic regions and associate genes to peaks. g:Profiler [64] was used to perform gene ontology analysis of genes associated with co-bound peaks and ggplot was used to create dotplots. Sequencing data files were submitted to the GEO repository (GSE240816).

## RNA extraction

Total RNA was extracted from frozen, crushed myometrium tissue samples, obtained from individuals at TNIL or TL (n = 6/group), using a Trizol/Chloroform extraction protocol. RNA samples were column-purified using RNeasy Mini Kit (Qiagen), treated with DNase I (Qiagen) to remove genomic DNA contamination, and subsequently subjected to standard Illumina Stranded Total RNA Prep Ligation with Ribo-Zero Plus protocol for paired-end sequencing.

## RNA-Seq data analysis

Raw data of FASTQ format were quality assessed and trimmed using fastp [58] followed by mapping to the hg38 genome using STAR [59]. Reads were quantified using featureCounts (exon reads) [65] and intron reads were quantified using SeqMonk's active transcription quantitation pipeline. Alternative transcript counts were summed together for every gene. Intron reads were then imported into DESeq2 for differential expression analysis [66]. Genes with a $\log_2$ fold change ≥ |1.5| and adjusted *p*-value ≤ 0.01 were considered significantly different. DESeq2 was used to assess the differential expression and data were plotted using the EnhancedVolcano [67] and the ComplexHeatmap [68,69] package in R, for heatmaps and volcano plots, respectively. The list of differentially expressed genes were used as input for Gene Ontology (GO) analysis with a threshold of adjusted *p*-value ≤ 0.05 (Benjamini-Hochberg), using the "enrichGO" function in the R package "clusterProfiler" [70,71]. Enrichment results were sorted by the adjusted *p*-value and grouped by the biological process. Deep-Tools (bamCoverage) [72] was used to generate bedgraphs which were visualized on UCSC. Sequencing data files were submitted to the GEO repository (GSE244863).

Genomic regions for enhancer RNA (eRNA) expression analysis were selected based on intergenic regions containing JUND peaks (common at both timepoints) and H3K27ac peaks at labor. Peak intersection was conducted via bedtools [73]. Read counts were obtained at these regions using featureCounts [65] and differential RNA expression analysis was

done using DESeq2 [66]. Regions with an eRNA $\log_2$ fold change of ≥ |1| and adjusted *p*-value ≤ 0.05 were considered as peaks with significantly changing signal intensity from TNIL to TL and differential eRNA expression was visualized using EnhancedVolcano [67] package.

**Motif enrichment analyses**

Enrichment of transcription factor binding motifs at regions exhibiting significant up-regulation of eRNA expression and transcripts was performed using "findMotifsGenome.pl" and "findMotifs.pl", respectively, from the HOMER [10] database. Significantly enriched motifs were calculated with *p*-value < 0.01.

## Supporting information

**S1 Fig. RNA-seq samples cluster based on laboring status at the time of tissue collection.** Principal component analysis (PCA) of RNA-seq samples in non-laboring tissues (TNIL) and laboring tissues (TL).
(PDF)

**S2 Fig. JUND ChIP-seq samples cluster based on laboring status at the time of tissue collection.** (A) Hierarchical clustering of JUND ChIP-seq samples from non-laboring (TNIL) and laboring (TL) tissues. Darker color indicates increased correlation. (B) Principal component analysis (PCA) of JUND ChIP-seq samples from non-laboring (TNIL) and laboring (TL) tissues.
(PDF)

**S1 Table. Genome-wide exon read-based RNA expression analysis in TNIL and TL myometrium.**
(XLSX)

**S2 Table. Comparison of RNA-seq datasets in human myometrium.**
(XLSX)

**S3 Table. Genome-wide intron read-based RNA expression analysis in TNIL and TL myometrium.**
(XLSX)

**S4 Table. Motif enrichment values at promoters of genes with increased expression at TL.**
(XLSX)

**S5 Table. JUND ChIP-seq differential binding analysis.**
(XLSX)

**S6 Table. Motif enrichment values at regions with JUND and PR binding.**
(XLSX)

**S7 Table. List of genes whose promoters are bound by JUND and PR and the association with TNIL/TL expressed genes.**
(XLSX)

**S8 Table. GO enrichment of JUND and PR bound promoters with increased expression at TL.**
(XLSX)

**S9 Table. JUND bound H3K27ac regions displaying differential eRNA levels.**
(XLSX)

**S10 Table. Motif enrichment values at JUND bound H3K27ac regions depicting increased eRNA reads at TL.**
(XLSX)

## Acknowledgments

The authors thank the donors, and the Research Centre for Women's and Infants' Health BioBank at the Lunenfeld-Tanenbaum Research Institute, and the Department of Obstetrics & Gynecology, at Sinai Health System particularly the staff of the Labor and Delivery Unit for their support for the human specimen's collection used in this study.

## Author contributions

**Conceptualization:** Nawrah Khader, Jennifer A. Mitchell.

**Data curation:** Nawrah Khader, Anna Dorogin, Oksana Shynlova.

**Formal analysis:** Nawrah Khader.

**Funding acquisition:** Jennifer A. Mitchell.

**Investigation:** Nawrah Khader.

**Methodology:** Nawrah Khader.

**Project administration:** Oksana Shynlova, Jennifer A. Mitchell.

**Resources:** Oksana Shynlova, Jennifer A. Mitchell.

**Software:** Nawrah Khader.

**Supervision:** Jennifer A. Mitchell.

**Validation:** Nawrah Khader.

**Visualization:** Nawrah Khader, Jennifer A. Mitchell.

**Writing – original draft:** Nawrah Khader.

**Writing – review & editing:** Nawrah Khader, Oksana Shynlova, Jennifer A. Mitchell.

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
