## [Decision Letter · Decision Letter 0]

PGENETICS-D-24-00402

JUND plays a genome-wide role in the quiescent to contractile switch in the pregnant human myometrium

PLOS Genetics

Dear Dr. Mitchell,

Thank you for submitting your manuscript to PLOS Genetics. After careful consideration, we feel that it has merit but does not fully meet PLOS Genetics's publication criteria as it currently stands. Therefore, we invite you to submit a revised version of the manuscript that addresses the points raised during the review process.

Please submit your revised manuscript within 30 days Mar 25 2025 11:59PM. If you will need more time than this to complete your revisions, please reply to this message or contact the journal office at plosgenetics@plos.org. Please include the following items when submitting your revised manuscript:

We look forward to receiving your revised manuscript.

Kind regards,

Paula E. Cohen

Section Editor

PLOS Genetics

Aimée Dudley

Editor-in-Chief

PLOS Genetics

Anne Goriely

Editor-in-Chief

PLOS Genetics

**Journal Requirements:**

1) Please provide an Author Summary. This should appear in your manuscript between the Abstract (if applicable) and the Introduction, and should be 150-200 words long. The aim should be to make your findings accessible to a wide audience that includes both scientists and non-scientists. Sample summaries can be found on our website under Submission Guidelines:  

https://journals.plos.org/plosgenetics/s/submission-guidelines#loc-parts-of-a-submission 

3) Some material included in your submission may be copyrighted. According to PLOSu2019s copyright policy, authors who use figures or other material (e.g., graphics, clipart, maps) from another author or copyright holder must demonstrate or obtain permission to publish this material under the Creative Commons Attribution 4.0 International (CC BY 4.0) License used by PLOS journals. Please closely review the details of PLOSu2019s copyright requirements here: PLOS Licenses and Copyright. If you need to request permissions from a copyright holder, you may use PLOS's Copyright Content Permission form. 

Potential Copyright Issues: 

i) Figures 1A, 3A, 6: Please confirm whether you drew the images / clip-art within the figure panels by hand. If you did not draw the images, please provide (a) a link to the source of the images or icons and their license / terms of use; or (b) written permission from the copyright holder to publish the images or icons under our CC BY 4.0 license. Alternatively, you may replace the images with open source alternatives. See these open source resources you may use to replace images / clip-art: - https://commons.wikimedia.org - https://openclipart.org/.  

Note: If your figures were made with Biorender, which aligns with PLOS publishing policies, please add this information in the figures legends.

4) Thank you for stating that "All sequencing data is submitted to GEO (https://www.ncbi.nlm.nih.gov/geo/), accession GSE240816." We strongly recommend all authors deposit their data before acceptance, as the process can be lengthy and hold up publication timelines. Please note that, though access restrictions are acceptable now, your entire minimal dataset will need to be made freely accessible if your manuscript is accepted for publication. This policy applies to all data except where public deposition would breach compliance with the protocol approved by your research ethics board. If you are unable to adhere to our open data policy, please kindly revise your statement to explain your reasoning and we will seek the editor's input on an exemption. 

5) Please upload the Supplemental Tables in a correct numerical order in the online submission form. Please ensure to refer to them in the Supporting Information legends as S1 Table and S2 Table, and so forth.  

**Reviewers' comments:**

Reviewer's Responses to Questions

**Comments to the Authors:**

**Please note that one of the reviews is uploaded as an attachment.**

Reviewer #1: JUND plays a genome-wide role in the quiescent to contractile switch in the pregnant human myometrium

Khader et al., PLoS Genetics 2024

In the manuscript Khader et al, the non-laboring and laboring states of human myometrial samples was compared. They examined gene expression changes and the mechanisms driving these. They present a model in which JunD homodimers are bound at both enhancers and promoters in both non-laboring and laboring states with the Progesterone Receptor (PR) recruited by JunD at promoters. Their model proposes that an increase in Fos expression at the onset of labor results in JunD/Fos heterodimers enhancing gene expression.

Overall, this is an interesting paper and looks at an important biology system which is not highly studied. The quality of the data is high and overall the analysis is well done. This is relevant since the complexity of looking at human tissues can be confounding. The comparison of exon and intron data was excellent to drive the argument for transcriptional regulation rather than other aspects of gene regulation. Below are some comments:

1. The authors state at line 120 “Principal component analysis (PCA) of all samples revealed that replicates belonging to the same group cluster together and show similar expression trends (S1 Fig).”

I found this statement confusing since on the PCA, TL 2 and TL 3 seem to cluster with the TNIL. Some better description of the PCA would be more helpful. The samples are what they are, but does it suggest that TL is highly variable? It may also be that the diagnosis of active labor may be variable since the samples were collected by caesarean section.

2. Related to the above, gene expression analysis was done with the aggregated data. It would be useful to see the variability in the samples (could be box plots in supplemental?), for example, for Fos. If there is variability, is TL2 and TL3 outliers?

3. The ChIP-seq data is very nice. Which samples were used for the ChIP-seq? Are they the same as the gene expression? Which ones?

4. In the motif analysis, JunB keeps coming up as a candidate. While I understand that motif searches are not generally specific to a family member, it did make me wonder about other family members, particularly since JunB can function as a repressor. Is the expression of other Jun family members altered? While JunD has been explored before, a better justification for following up on JunD may help clarify the question.

Minor:

In line 522, the authors say for ChIP-seq they use “TNIL or TL (n = 2, each)”. However, in the figures, they have 3 and 2. In that regard, are these samples the same as the ones for the gene expression? If so, which ones were used?

Figure S2, uses label LAB and TNIL rather than TL and TNIL. It would be good to be consistent.

Reviewer #2: This is a strong study that addresses the epigenetic and transcriptomic transition experienced by the human myometrium as it transitions into labor. It performs a novel JUND ChIP-Seq analysis on clinical specimens and integrates this with an RNA-Seq analysis. Its main conclusion is that JUND shows dynamic binding to the genome during the TNIL to TL transition, with some loci gaining JUND binding, some losing JUND binding and some retaining similar levels of JUND binding. This latter group is interesting because these sites are also bound by the progesterone receptor (PR) in both TNIL and TL specimens. I have several requests:

1) The 88 genes that are upregulated across this data set plus the two other data sets referenced (Table S2) should be discussed further. Inspection of these genes reveals a large number of inflammation-associated genes, which is consistent with prior studies and becomes very clear when the list is sorted high to low, and this should be acknowledged. Importantly, many of these genes are highly specific neutrophil chemoattractants or neutrophil growth factors (CXCL1, CXCL2, CSF3) or neutrophil markers (CXCR2, CSF3R, S100A8, S100A9, IL1B). The detection of these genes is consistent with the known influx of neutrophils into the laboring myometrium, but the presence of these cells (and thus many gene products in the RNA-Seq data) should be mentioned as a limitation of the study. More generally, there should be discussions of the fact that the ChIP-Seq and RNA-Seq experiments were performed on mixtures of cell types, how this fact influences interpretations, and what cell type is the most prevalent and so primarily accounts for the observed changes. Are there any data on the myocyte:non-myocyte ratio in the term myometrium that would help tease this out? Could those loci with increased JUND binding at TL, being ones that are enriched for inflammatory targets, represent JUND binding in influxing or activated immune cells? This possibility is consistent with the fact that immune cells do not express PR and these regions do not show PR co-occupancy. Since presumably the authors do not intend to study neutrophils, the motif enrichment shown in Figure 1F should also be repeated without the genes known to be expressed relatively specifically by neutrophils. This might increase the significance of the enrichments detected. Published single cell RNA-Seq data set of the human myometrium could be used to parse out which genes are expressed by various kinds of immune cells more generally.

2) It is unclear why PTGS2, IL6, CXCL2 and IL1B are singled out for special mention in the text and why certain genes are called out by name in Figure 1C.

3) There are n=3 TNIL specimens but only n=2 TL specimens for the JUND ChIP-Seq. Can the authors comment upon whether this might have caused the skewing towards higher gene expression in the TNIL specimens?

4) Figure 1B, the lateral wisps of data points look artifactual (for the TNIL side of the plot, this is the contiguous “line” of data points that extend out to -7.5 log2fc and -7.5 -log10P; see attached screenshot). I wonder whether these peaks are small and so should be excluded? Is there anything special about them?

5) The description of the JUND differential peak analysis is confusing (lines 185-195, Figures 2A and 2B). It starts off with what appears to be a tally of the absolute presence or absence of a peak, followed by a tally of peaks that show quantitatively different levels. If so, this should be made more explicit. But if I’m reading it correctly, I find this approach unnecessarily complex. How is it biologically meaningful to distinguish between whether a peak goes from present to “absent,” versus present to very low? Moreover, the key thing later on is to show that peaks with little change between TNIL and TL are the ones near PR-binding peaks. The authors should consider just dividing the peaks into three groups based upon cutoffs they choose for increasing and decreasing from TNIL to TL, and show a continuous plot in Figure 3C that demarcates where this cutoff is (and Figure 3C should be moved to Figure 2). This is the approach that was taken in the Dotts paper they cite, which I find more intuitive. Additionally, it’s confusing to claim that there is a “TNIL”-specific set of peaks when the top section of Figure 3C shows that many of these peaks are still there in the TL specimens.

5) For Figure 2D, please perform a GO analysis of the shared peaks. This is particularly important since these regions are the ones co-occupied by PR.

6) Lines 203-207. I do not understand this sentence, and the lack of details in the S3 legend adds to my confusion. Are these just upregulated genes? Additionally, how can it be that S3 claims no difference whereas the similarly minor peaks differences in the shared categories of Fig 2D are discussed as though there were real?

7) The choice to display data for ACTC1 and CCL24 in Figure 2E seems random and hence cherry-picked. Are these gene just examples of many that show the illustrated behavior? If so, what is the set of such genes and why not show more classic contraction associated genes instead of ACTC1?

8) Although touched upon in the Dotts paper, which provided the PR ChIP-Seq data used here, the idea that PR binding would be associated with genes that are upregulated with the transition to labor is novel and contracts the prevailing idea in the field that labor onset in humans is associated with “functional progesterone withdrawal,” meaning that there is a loss of PR function, presumably associated with decreased PR binding to DNA, despite maintenance of high serum P4 levels. Not much in vivo data supports this idea however. So the present study’s description of continued PR binding to genes that are upregulated in the transition to labor (and very few genes with PR binding that are downregulated) is important, and I believe the paper would be enhanced if this was acknowledged and perhaps emphasized in its own right.

Minor:

1) The text in Figure S4A is unreadable. An excel file would be better.

2) Figure S4B lacks an x-axis label. Also, I don’t understand why dots are plotted that seem to have Padj>0.05. It seems like some have Padj approaching 0.2 (Intracellular pH reduction), which means the enrichment is not significant and so should not be plotted. The same issue applies to Figure 5E.

3) Please clarify how Figure 1B differs from 1C. The text reads as though these are different data, not just different representations of the same data. Perhaps there is some subtlety regarding total transcripts and exon reads, but if so this should be made more clearly.

4) The redundant labeling of Fig 1B and 2C creates confusion. Why not just label the top of each heatmap with the orange and red bars, labeled respectively as TNIL and TL, and remove the orange/TNIL and red/TL legend labels on the side?

5) What are the small histograms in the upper left of Fig 2C and S2A?

6) Line 151 – “74 genes” – should this be Supplemental Table 3 (not 2)? Additionally, I don’t see any list that has only 74 genes on it.

7) The dot legend at the bottom of Fig. 1B has a grey category not present in the actual figure.

8) The words “enriched” and “enrichment” are used in multiple different ways throughout the manuscript, and this reduces readability. Sometimes they refer to the statistical enrichments seen with a GO analysis, sometimes to the enrichment of binding at a particular locus seen with ChIP-Seq. I would suggest changing the second usage to peak “concentrations” or “intensities.” and “enrichment plots” to “peak profile plots”, to avoid confusion.

9) The sentence in line 247-248 makes it sound like there are new data here that “confirm” the peak categories defined previously. But this was the analysis defined these categories in the first place, no? As above, I suggest moving Fig 3C to Figure 2 to improve clarity.

10) Line 249: “showed reduced absolute enrichment levels compared to TNIL ” – isn’t this observation that defined them as TNIL-specific peaks to begin with?

11) Figure S4B should be a main figure given its centrality to the main theme of the manuscript.

12) Line 556: Please expound on the IDR approach to determining the combined peak set.

Reviewer #3: In this study the authors identify differentially regulated genes expressed in human myometrial samples taken from term not in labor and term labor pregnancy. They also measure JunD, PR occupancy on promoters and correlated expression with transcription factor binding and identify JunD as a factor in changes in transcriptional pattern from TNIL to TL.

Sup Figure 1 line 119

The authors state “Principal component analysis (PCA) of all samples revealed that replicates belonging to the same group cluster together and show similar expression trends (S1 Fig).” Although the TNIL samples are clustered the TL samples are quite different and two are overlapping with the TNIL samples. This can also be seen in Fig 1b. Because of this I think that the TL samples can not be considered a cluster and the language used to describe them should be changed. Are there any clinically relevant differences between the TL samples that could be contributing to the variability?

Fig1F lines 150, 162.

In figure 1F the authors compare genes differently increased in TL based on intron reads to TL genes identified in other data sets. Since Exon reads were also measured within the same experiment as the Intron reads these two groups should be compared. They later state that this comparison was done and that AP-1 motifs were enriched but do not show the data.

Fig4 and 5

When comparing JunD peaks and PR and H3k27ac peaks why were only the shared TNIL and TL JunD peaks used in Fig 4 and 5? Based on the data from Fig 3 there are likely to be few hits, but they could be interesting and many reveal further patters that could better explain the regulation of expression of other genes such as CCL24 from Fig 2E. Also, does figure S4 include all JunD peaks or only the TNIL-TL shared peaks? Please clarify in the text.

Discussion and Results

The authors cite a mix of mouse and human data supporting their findings on the importance of PR in TL gene expression. In mice a decrease in progesterone occurs at the end of term and induces labor initiation. Since this does not occur in humans the authors should discus what impact their findings have on understanding the differences in PR dynamics in human and mouse labor.

**Have all data underlying the figures and results presented in the manuscript been provided?**

Reviewer #1: Yes

Reviewer #2: Yes

Reviewer #3: Yes

PLOS authors have the option to publish the peer review history of their article (what does this mean? ). If published, this will include your full peer review and any attached files.

**Do you want your identity to be public for this peer review?** For information about this choice, including consent withdrawal, please see our Privacy Policy .

Reviewer #1: No

Reviewer #2: No

Reviewer #3: No

**Figure resubmission:**
---

## [Editor Report · Decision Letter 1]

PGENETICS-D-24-00402R1

JUND plays a genome-wide role in the quiescent to contractile switch in the pregnant human myometrium

PLOS Genetics

Dear Dr. Mitchell,

Thank you for submitting your manuscript to PLOS Genetics. After careful consideration, we feel that it has merit but does not fully meet PLOS Genetics's publication criteria as it currently stands. Therefore, we invite you to submit a revised version of the manuscript that addresses the points raised during the review process.

Please submit your revised manuscript within 30 days May 01 2025 11:59PM. If you will need more time than this to complete your revisions, please reply to this message or contact the journal office at plosgenetics@plos.org. Please include the following items when submitting your revised manuscript:

We look forward to receiving your revised manuscript.

Kind regards,

Paula E. Cohen

Section Editor

PLOS Genetics

Paula Cohen

Section Editor

PLOS Genetics

Aimée Dudley

Editor-in-Chief

PLOS Genetics

Anne Goriely

Editor-in-Chief

PLOS Genetics

**Reviewers' comments:**

**Figure resubmission:**
---

## [Decision Letter · Decision Letter 2]

Dear Dr Mitchell,

We are pleased to inform you that your manuscript entitled "JUND plays a genome-wide role in the quiescent to contractile switch in the pregnant human myometrium" has been editorially accepted for publication in PLOS Genetics. Congratulations!

Before your submission can be formally accepted and sent to production you will need to address the minor comments provided by the reviewers, particularly Reviewer 2. In addition, please complete our formatting changes, which you will receive in a follow up email. Please be aware that it may take several days for you to receive this email; during this time no action is required by you. Please note: the accept date on your published article will reflect the date of this provisional acceptance, but your manuscript will not be scheduled for publication until the required changes have been made.

Yours sincerely,

Paula Cohen

Section Editor

PLOS Genetics

Aimée Dudley

Editor-in-Chief

PLOS Genetics

Anne Goriely

Editor-in-Chief

PLOS Genetics

Comments from the reviewers (if applicable):

Reviewer's Responses to Questions

**Comments to the Authors:**

Reviewer #2: The manuscript by Khader et al. is substantially improved. I have only a few remaining minor concerns:

1. line 226: Shouldn’t this call out Figure 2B as well? 2C only shows the peaks, across replicates, that are increased in TNIL, not the observation that there are more such peaks in TNIL compared to TL.

2. Line 357: Please add a phrase that describes how the data were “filtered”.

3. Line 370: REL should be RELA

4. Line 441: “Contraction-related” is not quite right since CXCL8 is not contraction-related. Maybe “labor-associated” instead?

5. Line 477: please change “implicates” to “involves”. I think this what the meaning is intended to be.

Reviewer #3: The authors have addressed all of my concerns.

**Have all data underlying the figures and results presented in the manuscript been provided?**

Reviewer #2: Yes

Reviewer #3: Yes

PLOS authors have the option to publish the peer review history of their article (what does this mean? ). If published, this will include your full peer review and any attached files.

**Do you want your identity to be public for this peer review?** For information about this choice, including consent withdrawal, please see our Privacy Policy .

Reviewer #2: No

Reviewer #3: No

**Data Deposition**

http://datadryad.org/submit?journalID=pgenetics&manu=PGENETICS-D-24-00402R2

**Press Queries**

---

## [Editor Report · Acceptance letter]

PGENETICS-D-24-00402R2

JUND plays a genome-wide role in the quiescent to contractile switch in the pregnant human myometrium

Dear Dr Mitchell,

We are pleased to inform you that your manuscript entitled "JUND plays a genome-wide role in the quiescent to contractile switch in the pregnant human myometrium" has been formally accepted for publication in PLOS Genetics! Your manuscript is now with our production department and you will be notified of the publication date in due course.

With kind regards,

Anita Estes

PLOS Genetics

On behalf of:
